# Seed Viability and Potential Germination Rate of Nine Endemic *Boswellia* Taxa (Burseraceae) from Socotra Island (Yemen)

**DOI:** 10.3390/plants11111418

**Published:** 2022-05-26

**Authors:** Salem Hamdiah, Lukáš Karas, Kateřina Houšková, Kay Van Damme, Fabio Attorre, Petr Vahalík, Hana Habrová, Samuel Lvončík, Klemen Eler, Petr Maděra

**Affiliations:** 1Department of Forest Botany, Dendrology and Geobiocoenology, Faculty of Forestry and Wood Technology, Mendel University in Brno, Zemědělská 3, 613 00 Brno, Czech Republic; balagahar@gmail.com (S.H.); kay.vandamme@ugent.be (K.V.D.); hana.habrova@mendelu.cz (H.H.); samuel.lvoncik@mendelu.cz (S.L.); petr.madera@mendelu.cz (P.M.); 2Biotechnical Faculty, University of Ljubljana, Jamnikarjeva 101, 1000 Ljubljana, Slovenia; klemen.eler@bf.uni-lj.si; 3Department of Silviculture, Faculty of Forestry and Wood Technology, Mendel University in Brno, Zemědělská 3, 613 00 Brno, Czech Republic; katerina.houskova@mendelu.cz; 4Centre for Academic Heritage and Archives & Ghent University Botanical Garden, Ghent University, K.L. Ledeganckstraat 35, 9000 Ghent, Belgium; 5Department of Environmental Biology, Sapienza-University of Rome, Piazzale Aldo Moro 5, 00185 Rome, Italy; fabio.attorre@uniroma1.it; 6Department of Forest Management and Geoinformatics, Faculty of Forestry and Wood Technology, Mendel University in Brno, Zemědělská 3, 613 00 Brno, Czech Republic; petr.vahalik@mendelu.cz

**Keywords:** *Boswellia*, endangered species, Frankincense, propagation, Soqotra

## Abstract

The endemic *Boswellia* species (Burseraceae) on Socotra Island (Yemen) are of great local significance due to their various local ethnobotanical uses. However, despite the fact that these trees are endangered, little is known about their biology. We tested seed germination rates in controlled experiments (trials of 21 days) for two subsequent years and for nine endemic taxa of *Boswellia* occurring on Socotra Island. For this, seeds were collected island-wide from a wide range of localities and for several populations per species. We observed differences in germination among *Boswellia* species, among species and localities and among both years, which indicates that the development of seeds is strongly affected by external ecological factors. Although we noted a large variation in seed germination (relatively high in *Boswellia socotrana*), and half of the species showed relatively low mean daily germination, our study indicated that all endangered endemic Frankincense Tree taxa of Socotra harbor the potential for in situ conservation through recruitment, given that known impacts can be reduced in local replantation areas (e.g., grazing).

## 1. Introduction

All Frankincense tree taxa, or members of the genus *Boswellia* (Burseraceae), are globally endangered, being harvested intensively for their oleo-gum resin [1]. The unsustainable use of olibanum or frankincense leads to the overexploitation of *Boswellia* populations on a global scale [1,2]. In addition, *Boswellia* trees are endangered because of overgrazing by livestock, which is considered an important negative factor influencing natural regeneration [1,3]. This is particularly problematic in arid regions, where the majority of these trees reside. For example, in Eritrea, human-induced factors such as land clearing for agriculture, overgrazing and overtapping of resin are threatening *Boswellia papyrifera* (Del.) Hochst populations [4]. *Boswellia papyrifera* in central and eastern Africa is declining at an alarming rate, as natural populations are facing degradation due to agricultural expansion, overgrazing, fires, poor incense harvesting practices, shifting cultivation, termites and other infestations; urgent conservation measures are required to save the species [5]. Similar factors are affecting other species in Africa and the Arabian Peninsula [6,7]. In general, based on extensive research throughout the geographic range of the five most economically important *Boswellia* species, Bongers et al. [1] concluded that *Boswellia* populations are globally threatened by over-exploitation and ecosystem degradation, jeopardizing future resin production. These changes are caused globally by increased human population pressure on *Boswellia* woodlands mainly through livestock grazing and unsustainable tapping.

As a result of mainly human impacts, frankincense tree populations are declining rapidly in the wild. However, studies about population decline and the relationships with potential causes are relatively few. The most comprehensive study revealed evidence of the population collapse of *B. papyrifera* throughout its geographic range in Africa [1]. Using inventories of 23 populations consisting of 21,786 trees, growth-ring data from 202 trees and demographic models based on 7246 trees, the latter authors found that over 75% of the studied populations lacked young trees, and natural regeneration had been absent for decades; therefore, the projected frankincense production could be halved in 20 years [1]. Additionally, a population structure analysis of *B. papyrifera* in Eritrea made by Ogbazghi et al. [4] shows that there seems to be an overall absence of juvenile trees. Natural regeneration was found only in two areas in which trees were not tapped for resin and that were inaccessible to livestock. On Socotra Island (Yemen), *Boswellia* population studies are nearly lacking, with two exceptions. Attorre et al. [8] evaluated all *Boswellia* species known at that time on Socotra Island and found natural regeneration in only two cliff-rooted species (*Boswellia popoviana* and *B. dioscoridis*). Lvončík et al. [9] concentrated on the evaluation of the long-term development of the largest population of *Boswellia elongata* in Homhil Nature Sanctuary, also showing an overall lack of juvenile trees and a strong effect of global climate change, in this case, unusually strong cyclones.

Socotra Archipelago (Yemen), situated in the northwestern Indian Ocean, has the highest diversity of Burseraceae species in the world [10]. Eleven taxa of *Boswellia* can be found there, all of them are endemic, and five species of *Commiphora*, four of which are endemic [10,11,12]. Frankincense trees in Socotra have numerous environmental, socio-economic, traditional medicine and potentially industrial benefits as the ethnobotanical knowledge indicate [10,11] which, besides grazing, all add to the increased vulnerability and human use of these species. Finally, global climate change contributes in both direct and indirect ways to a rapid population decline of frankincense trees on Socotra Island [9] and is estimated to have an impact on *Boswellia* globally [1]. Therefore, renewed efforts should be made to improve our knowledge of particularly range-restricted insular populations of endangered *Boswellia* such as those on Socotra Island, in order to strategize aimed future conservation actions. A better understanding of the biology of these species can be a powerful tool in local long-term conservation. As a comparative example in Socotra, studies on the biology of the endemic Socotran Dragon’s Blood Tree *Dracaena cinnabari* Balf. f., are useful tools to understand the potential and limitations in conservation activities for the entire group of Dragon Trees in the world [13].

Scientific studies on the biology, including seed germination ability of *Boswellia* species from Socotra Island, as a basis of natural regeneration, and the potential intrinsic factors for the decline, are missing. In other Frankincense tree taxa, relatively low germination rates were recorded. Swartout and Solowey [14] mentioned significantly low germination viability for *Boswellia sacra* Flueck. (4–16%) in Israel, and significantly low germination rates (up to 8%) in this species were also recorded by Eslamieh [15]. The viability of seeds of *Boswellia papyrifera* in Ethiopia was estimated at ca. 49% to 59% [16]. Higher germination rates were found in the latter study in untapped trees, thus tapping (i.e., manual extraction of resin through administering wounds to the tree) may negatively affect seed viability. Rijkeers et al. [17] published similar results in *Boswellia papyrifera* in Ethiopia, where untapped trees produced three times as many healthy, and properly filled germinable seeds as tapped trees. Germination success was highest in stands with untapped trees (>80%) and lowest with those for tapped trees (<16%). Experiments that have included treatments for breaking seed dormancy were carried out for African *Boswellia* species [16,17,18,19]. Such experiments are key to future replantation of these endangered tree species for conservation efforts, and to assess the effects of human impacts on tree biology.

The general lack of natural regeneration of in situ populations of *Boswellia* [1] could be caused by both extrinsic (e.g., grazing, climate change) and intrinsic factors (e.g., floral biology, physiology). Our focus in this study is on assessing the influence of intrinsic biological factors related to generative reproduction (seed germination), in the function of future conservation of these endangered species. The main question of the current study is to establish whether seed germination, as a major intrinsic factor, is a limiting factor for the natural regeneration of *Boswellia* species on Socotra Island. Thus, we assessed the germination rates and germination energy of seeds of nine *Boswellia* taxa (*Boswellia ameero* Balf. f., *Boswellia bullata* Thulin, *Boswellia dioscoridis*, *Boswellia elongata*, *Boswellia nana* Hepper, *Boswellia popoviana*, *Boswellia scopulorum* Thulin, *Boswellia socotrana* subsp. *aspleniifolia* (Engl.) Lvončík, and *Boswellia socotrana* Balf. f. subsp. *socotrana*) occurring on Socotra Island through seed germination trials under controlled conditions, taking into account potential variation of germination between different years and different populations.

## 2. Results

### 2.1. Inter-Locality Variabilities of Different Boswellia Taxa in 2020 and 2021

The highest proportion of germinated and fresh seeds (ca. 90%) and the lowest proportion of empty seeds (up to 1%) was observed in *B. socotrana* in both subspecies and tested sites in 2020 (subsp. *socotrana* BSSA 20 from Ayhaft and subsp. *aspleniifolia* BSAS from Shata Qalansiyah; Figure 1). The other tested *Boswellia* species showed a lower proportion of germinated and fresh seeds (up to 40%) and a higher proportion of empty seeds (above 40%). Eight out of eleven localities had more than 50% empty seeds in the 2020 seed sets. The lowest proportion of germinated and fresh seeds (up to 5%) was found in *B. bullata* BBT from Taida’ah and *B. elongata* BEM from Makalihim, no germinated or fresh seeds were found for *B. elongata* BEDB from Diburak and *B. popoviana* BPK from Klisan; the seeds of these species were mostly empty. Most samples from 2020 had 10% or less of dead seeds or seeds damaged by insects (Figure 1); only seeds of *B. bullata* BBT from Taida’ah and *B. elongata* BEH from Homhil contained ca. 30% of dead or insect infected seeds.

In 2021, the highest proportion of germinated and fresh seeds was observed in *B. socotrana* subsp. *socotrana*, *B. socotrana* subsp. *aspleniifolia* and *B. dioscorodis* reaching 70–80% (Figure 2), for both subspecies of *B. socotrana* values were lower than in 2020. For *B. socotrana* subsp. *aspleniifolia*, the high proportion of these seeds were similar at all tested localities (no statistical differences in germination rate, see Figure 3) but for *B. socotrana* subsp. *socotrana* (BSSZ), seeds only from Zarkan reached the highest value and seeds from Ayhaft (BSSA) and Homhil (BSSH) reached up to 40% of germinated and fresh seeds (statistical difference in germination rate, see Figure 3). Seeds of *B. bullata* contained ca. 35–50% germinated and fresh seeds at all localities, seeds of other species reached a similar proportion of these seeds only at one locality, i.e., *B. ameero* (BADN) from Danagahan, *B. elongata* (BEH) from Homhil, *B. nana* (BNH) from Hamadarah and *B. popoviana* (BPD) from Sheebhan; the seeds from other localities contained the minimal proportion of these seeds (usually up to 15%) and showed significantly lower germination rate. Seeds of *B. elongata*, *B. scopulorum* and *B. popoviana* at most localities were empty, and those of *B. ameero*, *B. bullata* and *B. dioscoridis* at most localities were dead or damaged by insects.

For the samples of 2021, *Boswellia socotrana* subsp. *aspleniifolia* reached the relatively highest germination rate of full seeds (ca. 70–90%) in all localities (Figure 4*). B. dioscoridis*, *B. elongata*, *B. popoviana* and *B. socotrana* subsp. *socotrana* showed a high germination rate of full seeds (ca. 70–95%) as well, but not in all localities (Figure 4). *B. dioscoridis* reached the highest values (ca. 90%) at Wadi Difa’araha (BDDF), statistically significantly lower values (ca. 70–80%) at Wadi Diasraha (BDDA) and Wadi Difgasfa (BDDG) and the lowest values (ca. 10–30%) at Halah (BDH) and Sheebhan (BDS). *B. elongata* reached the highest values (ca. 70–95%) at Homhil (BEH) and Shabarah (BES) and statistically significantly lower values (usually up to 50%) at Diburak (BEDB), Dishal (BEDS) and Makalihim (BEM). *B. popoviana* reached the highest values (ca. 85%) at Sheebhan (BPD) and Firmihin (BPF) and statistically significantly lower values (usually up to 50%) at Momi Falang (BPV) that did not differ from the zero germination rate of full seeds at Bedgofahar (BPB) and Klisan (BPK). *B. socotrana* subsp. *socotrana* reached the highest values (ca. 85%) at Zarkan (BSSZ) and statistically significantly lower values (ca. 50%) at Ayhaft (BSSA) and Homhil (BSSH). *B. scopulorum* from Bedgofahar (BSB) showed a high (ca. 80%) germination rate of full seeds too, but with high variability (from 22% to 100%). *B. ameero* did not reach such high values of germination rate of full seeds (maximum of ca. 65% at Qatariyah (BAQ), lower values of ca. 45% at Danagahan (BADN) and the lowest values up to ca. 20% at Ayhaft (BAA), Dixam (BADX) and Firmihin (BAF)). *B. bullata* has an average germination rate of full seeds (usually 40–60%) without significant differences between localities.

In 2021, *Boswellia socotrana* subsp. *aspleniifolia* seeds germinated relatively fast in all localities in comparison to other taxa (highest germination energy; Figure 5). There were no statistically significant differences in the germination energy of these seeds; within 4 days from the start of the germination test, most of the full seeds (65–85%) germinated. *B. nana* as well as *B. elongata* and *B.socotrana* subsp. *socotrana* also had a similar seed germination energy (above 50% on average), but not in all localities. Other species germinated more slowly, with less than 50% of full seeds germinating within 4 days of the start of the germination test. The lowest germination energy (up to 5%) was recorded in some localities for *B. ameero* (from Ayhaft BAA and Firmihin BAF), *B. dioscoridis* (from Wadi Diasraha BDDA, Wadi Difa’araha BDDF, Wadi Difgasfa BDDG and Halah BDH), *B. elongata* (from Diburak BEDB) and *B. popoviana* (from Bedgofahar BPB, Sheebhan BPD and Klisan BPK).

### 2.2. Average Inter-Annual and Inter-Species Variability of Germination Rates

Most of the species showed a similar germination rate between the two sampled years, but only three taxa (*Boswellia bullata*, *B. socotrana* subsp. *aspleniifolia* and *B. socotrana* subsp. *socotrana* at BBT, BSAS and BSSA) showed significant differences (Figure 6). *B. socotrana* subsp. *aspleniifolia* (BSAS) and *B. socotrana* subsp. *socotrana* (BSSA) showed relatively higher germination rates in 2020, *B. bullata* (BBT) in 2021.

When comparing the other species’ average germination rates, except *B. socotrana*, other *Boswellia* species show a relatively low germination rate, in half of the cases lower than 10%. It was usually due to the high proportion of empty seeds (see Figure 6 vs. Figure 7). In some cases, germination rates were 0% (BEDB, BPK).

### 2.3. Peak Value Indices

Seeds of all of the Socotran species started germinating in the first week of the germination test (Figure 8 and Figure 9). Both subspecies of *B. socotrana* started germinating during the first day of the test and *B. popoviana*, and *B. scopulorum* started germinating at the end of the first week. *B. ameero*, *B. bullata*, *B. elongata*, *B. nana*, *B. socotrana* subsp. *aspleniifolia* and *B*. *socotrana* subsp. *socotrana* germinated fast and reached peak values in the first week of the test, while the remaining species (*B. dioscoridis*, *B. popoviana* and *B. scopulorum*) in the second week. In 2020 and 2021, seeds of *B. socotrana* of both subspecies germinated the fastest (in 2021 closely followed by *B. nana*). All of the species finished germinating in the first or second week of the germination test, there were almost no germinating seeds found in the third week.

## 3. Discussion

### 3.1. Factors Affecting Germination of Boswellia Seeds

On average, we found germinated seeds in nine Socotran *Boswellia* taxa under controlled conditions. The observed germination rate among species and localities varied highly (*B. ameero* 0–70%, *B. bullata* 0–65%, *B. dioscoridis* 5–90%, *B. elongata* 0–45%, *B. nana* 30–70%, *B. popoviana* 0–65%, *B. scopulorum* 5–25%, *B. socotrana* subsp. *aspleniifolia* 25–94%, and *B. socotrana* subsp. *socotrana* 5–92%, for further details see Appendix A, Table A1 and Table A2). We did not have seeds of two species (*B. hesperia* sp. prov. and *B. samhaensis* Thulin and Scholte) occurring on the Socotra Archipelago available for the experiment. In 2022, after our visit to Samha Island, we found one fruiting tree of *B. samhaensis* and collected a limited number of seeds. A fast preliminary experiment of 20 seeds showed that after three days six seeds germinated, and on the fourth day, another one germinated as well. Therefore, *B. samhaensis* also has germinable seeds with a preliminary germination rate of 35%. There are only a few studies focusing on germination trials with seeds of *Boswellia* species, but generally, most of them report relatively low germination rates. Well known for a low germination rate is *Boswellia sacra*, varying between 4 and 16% [14,15]. The germination rate of *Boswellia papyrifera* was found to range from 4 to 7% under different treatments [20]. Higher germination rates were published by Eshete et al. [16] (40–72%) and by Rijkers et al. [18] (14–94%), but in both cases, the germination rate was counted from filled (full, viable) seeds. Similarly, Savithramma et al. [21] mentioned a 70% germination rate for *Boswellia ovalifoliata* N.P.Balakr. The last investigated species was *Boswellia dalzielii* Hutch. [17], associated with a germination rate in the field experiment (seeds were sown without any pre-treatment to the pots in the nursery) between 30 to 35%.

Tapping was found to be an important factor in decreasing the number of viable seeds and their germination rate [16,18]. Only naturally oozed resin is collected on Socotra Island, mostly without tapping [11]. Therefore, tapping is perhaps not the most important factor affecting the *Boswellia* trees’ vitality on Socotra and therefore may at this point not affect seed germination potential. Nevertheless, we did not systematically assess the tapping of the mother trees in our trials, as we focused on general seed germination in the first step.

Nonetheless, a high percentage of empty seeds was found across almost half of the taxa we included in our study (*Boswellia elongata*, *B. nana*, *B. scopulorum* and *B. popoviana*). Similar to our results, Rijkers et al. [18], Adam and El Tayeb [20] mentioned high percentages of empty seeds for *B. papyrifera* too. Additionally, Swartout and Solowey [14] reported only 4–17% of viable seeds of *Boswellia sacra* were detected by sink test. The high ratio of empty seeds could be caused by some external factors such as insufficient humidity or low soil nutrient content during fruit development. Climatic factors as a reason for low seed quality are mentioned by de Souza et al. [22]. Another explanation for a high proportion of empty seeds could be in hybridogenic origin within the evolution of these island species. Additionally, self-fertilization is mentioned as the main cause of empty seeds in *Fagus sylvatica* L. [23]. Fuentes and Shupp [24] demonstrated that empty seeds may reduce predation of full seeds of *Juniperus osteosperma* (Torr.) Little in Utah by seed-eating birds, therefore, a high ratio of empty *Boswellia* seeds on Socotra could also be a natural phenomenon.

Floral biology of two *Boswellia* species (*B. ovalifoliata* and *B. serrata* Roxb. Ex Colebr.) was investigated in detail by Raju et al. [25] and Sunnichan et al. [26], respectively. Strictly self-incompatible flowers were documented for both the above-mentioned species. With high probability, this will be the case for *Boswellia* species on Socotra as well. In populations with low tree density where individual specimens are far apart (which is often the case of *Boswellia* species on Socotra), most flowers could be self-pollinated and thus the fruit set is very low. Under open pollination, the fruit set was only about 10% of *Boswellia serrata* [26]. Unfortunately, we have not investigated this phenomenon on Socotra yet.

On the other hand, there is high importance on insects in the pollination of *Boswellia* species. Raju et al. [25] and Sunnichan et al. [26] report mainly bees as the principal pollinator insect. Sunbirds can serve as pollinators as well, but floral characteristics suggest that entomophily is the main mode [25]. The wind is not a vector for the spreading of pollen grains [26]. Raju et al. [25] found garden lizards to be predators of pollinating insects in *Boswellia ovalifoliata*, which can affect the pollination rate of this tree species. On Socotra, García and Vasconcelos [27] found a few species of geckos that may pollinate the dragon’s blood tree (*Dracaena cinnabari*), therefore, similarly they may serve as pollinators for *Boswellia* species.

Not many seeds were infested with insects in the Socotran *Boswellia* species (3% in 2020, 2% in 2021), which contradicts the figures for *Boswellia papyrifera* varying from 15.8 to 16.6% [16], 19.0–24.5% [18] to 55.7% [20]. Other predators of buds and flowers are possible, such as weevil or the palm squirrel, predating on fruits of other *Boswellia* [25].

We found high variability in germination rate between the years in three taxa (BBT, BSAS, BSSA) and also among different localities of the same *Boswellia* species on Socotra. This may indicate considerable sensitivity to the above-mentioned factors and there may be other factors that are not yet identified, affecting seed development. Over all these potential (yet undefined) factors, our results suggest that the problem of missing regeneration of *Boswellia* species on Socotra Island is not caused by internal biological factors related to generative reproduction. Although the germination rate is zero in one year, it is sufficient in another, with the exception of BEDB and BPK. We can conclude that individual trees of the different *Boswellia* taxa on Socotra are producing a sufficient amount of germinable seeds. Therefore, the problem of missing sapling recruitment is probably connected with external ecological factors acting on the growth of the seedlings after their germination.

### 3.2. Phenology of Fruit Ripening

We observed that the time of *Boswellia* capsules ripening on Socotra begins in the first half of May (*B. bullata*) and continues until the end second half of June (*B. popoviana*). However, there is a strong variability among the localities of individual species. None of the species were observed to ripen outside of May–June, with the exception of *B. dioscoridis*, which seems to ripen gradually during the whole year (Table 1).

The fruit ripening in *Boswellia* on Socotra (May–June) is potentially connected with the first rainfalls at the beginning of the summer monsoon, and subsequently, the winged seeds [25] are spread by strong summer winds [28]. However, it seems that most seeds fall in the proximity of the parental tree, a few meters to tens of meters [9]. The first natural seedlings appear in September with last summer monsoon rainfalls and their growth is supported by showers within the winter monsoon from October to December [9]. Additionally, high air humidity and horizontal precipitation in July and August [29] can substantially contribute to the survival of young seedlings. Similarly, *B. ovalifoliata* seed germination is connected with the first rainfalls [25].

### 3.3. Natural Regeneration

A relatively high recorded seed germination energy shows an excellent adaptation of seeds to the harsh dry environment with sporadic precipitation. The seeds are prepared to germinate immediately after the first rains at the beginning of the short rainy season. Similarly, other experiments have shown that seeds of *Boswellia papyrifera* start germinating around the fifth day after trial establishment [18,20]. In general, *Boswellia* seeds have short dormancy, after one year of storage, the germination rate may drop to half [16]. Swartout and Solowey [14] even report a lack of dormancy in *Boswellia* species which means that seeds of *Boswellia* species do not actually need pre-sowing treatment; the best solution is to soak them in cold water to germinate [20], as they likely cannot be stored at room temperatures indefinitely.

Although natural regeneration occurs in most *Boswellia* species on Socotra, the recruitment is missing, especially in ground-rooting species [8]. Their populations are thus gradually aging as Lvončík et al. [9] convincingly documented in the case study from Homhil (Socotra) for the *B. elongata* population. Similar population structure and development are documented across the entire area of *Boswellia* global distribution [1]. Missing or poor natural regeneration resulting in an unbalanced age structure of *Boswellia* populations is mentioned by many authors, for instance, for *B. papyrifera* in Sudan [20], in Eritrea [3,4,30] and in Ethiopia [6,31,32]. The healthy, regenerating populations of *B. papyrifera* in Ethiopia are unique [33]. Tolera et al. [34], investigating the population age structure of *B. papyrifera*, stated that the current populations in Ethiopia lack successful recruitment since 1955. This is in agreement with findings made by Lvončík et al. [9] for *B. elongata* on Socotra Island.

All the above-mentioned authors listed several factors to explain the missing young age classes of trees in *Boswellia* populations, more often intensive resin harvesting leading to extensive stem injuries and to decreasing of trees vitality that can be thereafter more easily affected by insect and fungal diseases; frequent human-induced fires; changes in land-use followed by land degradation or over-grazing. However, the fact that the success rate of seedling recruitment is highly limited due to water stress and nutrient-poor soils, must also be taken into account [25].

On the other hand, some authors mention the aspect of inaccessibility in species growing in rocky habitats (rock-dwellers). Ouédraogo and Thiombiano [17] proved this fact for *B. dalzielii* in Burkina Faso or Attorre et al. [8] for *B. popoviana* on Socotra Island. They attribute the success in natural regeneration to the inaccessible terrain for goats and to the remoteness of localities from inhabited areas.

An alternative way of regeneration could be vegetative propagation, as an adaptation to specific stress conditions such as wildfires or drought. Recruitment by root suckers is more common than sexual reproduction in *Boswellia dalzielii* [17]. Adam and Osman [35] proved the ability of *Boswellia papyrifera* stumps to create sprouts in high percentages which allow tree regeneration by coppicing. Propagation by cuttings with a high percentage of rooting was reported by Abiyu et al. [36] and Haile et al. [37], which may help overcome the population recruitment bottleneck of the *Boswellia* species [37] if properly managed and assisted to enhance the genetic diversity, including the directed propagation of hybrids and hybridogenic species that appear very often on Socotra [15,38,39] due to common occurrence of ten different species’ distinct characteristics.

## 4. Materials and Methods

### 4.1. Seed Collection and Preparation

We covered a significant area on Socotra Island with the aim to collect seeds of all known *Boswellia* species from different environments. The collection was carried out from April to June in two subsequent years (2020–2021), the first from May to June 2020 including 12 localities, the second end of April until the end of June 2021 covering 35 localities ranging between 1 and 932 m asl (Table 1, Figure 10). We aimed to collect seeds from five different localities for each *Boswellia* species (not possible for all species). Each species at a specific locality was assigned a code (Table 1) that is used in the graphs in the results. Seed collection and germination tests were part of the project “Conservation of the endangered endemic *Boswellia* trees on Socotra Island (Yemen)”, the project has written permission from the Environmental Protection Authority, Hadibo, Socotra (EPA), and EPA is a partner of the project and informed about the project activities.

Fruits were collected from at least ten adult trees per species, chosen randomly within each locality. Ripe fruits, recognized by capsules being reddish, black, or brown (not green), were collected directly from the trees and transported in separate recipients per species and locality using labeled cotton bags. The fruits were air-dried for two–three days, then the seeds were removed from the dried capsules manually and separated. The separated seeds were stored in labeled envelopes in the local *Boswellia* seed bank on Socotra (storage temperatures uncontrolled, ranging between 28 °C and 39 °C). The seeds were used for seed germination laboratory trials within three to maximally six months of collection.

### 4.2. Trials

A seed germination test was conducted for a random sample of 500 (in 2020) and 100 seeds (in 2021; due to lower amounts) from all collected species and localities. Seeds of each sample were divided into five replicates and germinated in Petri dishes. Conditions for the tests were according to the conditions for seed germination of most of the woody plants according to International Rules for Seed Testing [40] as follows: the dark phase for 16 h at 20 °C and the illuminated phase for 8 h at 30 °C. The trial was observed daily for 21 days straight during which germinated seeds were counted. A seed was deemed germinated when the radicle was at least as long as the size of the seed (Figure 11). All seed germination tests were carried out in Socotra. After each observation, seeds were transferred to a separate container daily to avoid double counting in the next day. After the end of the experiment, ungerminated seeds were cut and sorted into four classes: fresh, empty, dead, or damaged by insects, defined as follows:fresh—the tissue inside was solid, light or greenish, assuming start of germination in a longer time than the length of the germination trial;empty—without any tissue inside or remnants of tissue that filled less than half the internal space of the seed;dead—the tissue inside was soft, brownish and filled more than half the internal space of the seed;damaged by insects—seeds containing insects at different stages of development or an insect exit hole was visible in the seed coat.

Germination tests started at the beginning of October 2020 on seven *Boswellia* taxa endemic to Socotra Island from 12 localities collected in 2020; the next germination tests were conducted in September 2021 on 34 different seed sets belonging to nine taxa from the island, collected in 2021 (see all species and localities in Table 1).

### 4.3. Data Evaluation

The following parameters were selected for data processing and evaluation which are described below.

Germination rate (in %) was defined as the number of germinated seeds at the end of the experiment (after 21 days) out of the total number of seeds (including all seeds, full and empty) [41].

Germination rate of full seeds (in %) was defined as the number of germinated seeds at the end of the experiment (after 21 days) out of all full seeds. Full seeds are defined here as all seeds (germinated, fresh, dead, damaged by insect) that were determined as non-empty at the end of the germination test.

Germination energy (in %) was defined as the number of germinated seeds determined on day 4 after setting up the germination test from the total number of seeds [42,43] (including all seeds, full and empty).

Germination energy of full seeds (in %) was defined as the number of germinated seeds on day 4 after setting up the germination test from all full seeds.

Mean daily germination—Czabator [41] presented a peak value as a maximum quotient derived by dividing the number of germinants accumulated per day by the corresponding number of days, which is the mean daily germination of the most vigorous components of the seed lot. All values (not only peaks) are shown in the results (graphs).

Mean daily germination and percentages of germinated, fresh, dead, empty, and insect-damaged seeds were processed in MS Excel for the creation of graphs. We used Statistica 12.0 for data analysis and creation of the statistical graphs; one-way ANOVA was conducted to assess intra-locality variability (i.e., variability among localities of one species in the same year) of the data of the germination rate, the germination rate of full seeds and the germination energy of full seeds. Box plots show mean values, standard deviations and 95% confidence intervals. For inter-annual and inter-species variability, two-way ANOVA was used for the data of the germination rate and the germination rate of full seeds. Mean values are shown in graphs and the vertical bars represent 95% confidence intervals. When a significant difference was observed, mean separation was performed using Fisher’s least significant difference (LSD) test. All statistical tests were performed at a significance level of *p* < 0.05.

## 5. Conclusions

Our results are relevant to the conservation of the endangered *Boswellia* species in Socotra, as they indicate a realistic potential for seed germination for all species and for most localities. Potential human-mediated factors affecting significant differences between the years, between localities, or between species, are currently not examined in detail. However, there are initial indications that some species (e.g., *Boswellia socotrana*) show more potential for natural regeneration (under the conditions applied here, and without taking into account grazing impacts) than others. Mean daily germination remains relatively low for more than half of the species. Low germination rates in certain species or certain areas should be examined in more detail to understand the potential (human or other) impacts on their biology.

However, general factors impacting the Socotran terrestrial ecosystems are well understood, such as overgrazing and climate change impacts through recent cyclone effects [13,42,43]. In situ conservation and local seed germination in local nurseries on Socotra combined with replantation in nature (protected from browsing and replanted within the same area to avoid population mixing) are realistic additional strategies for improving natural protection and natural regeneration. As most Frankincense tree species in Socotra serve a variety of ecosystem services, among which their many ethnobotanical uses [11], more than one strategy for replantation is useful, besides protection of the natural terrestrial habitats, which are under various pressures. The current study on seed germination is a step towards strategizing as it improves the understanding of the future chances of success of local conservation endeavors.

## Figures and Tables

**Figure 1 plants-11-01418-f001:**
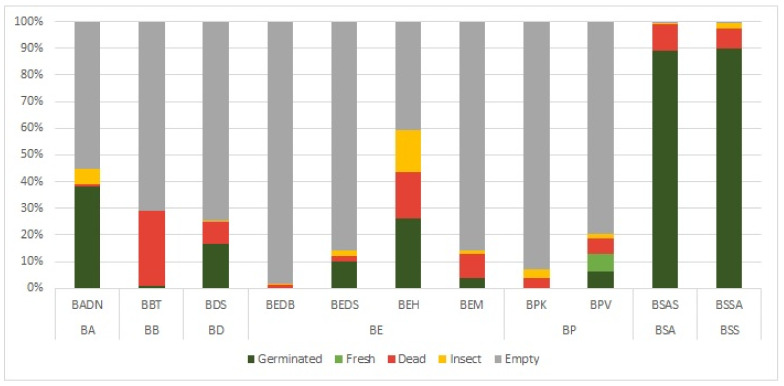
Seed category proportions of the seven *Boswellia* taxa collected from 12 localities (Socotra Island) in 2020 (for abbreviations, see Table 1).

**Figure 2 plants-11-01418-f002:**
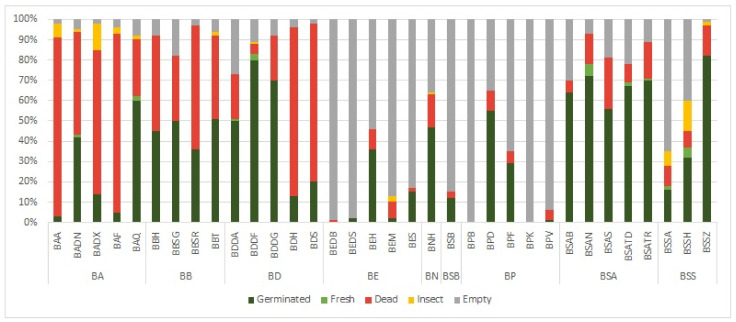
Seed categories proportion of all nine taxa of *Boswellia* collected from different localities on Socotra Island in 2021 (for abbreviations, see Table 1).

**Figure 3 plants-11-01418-f003:**
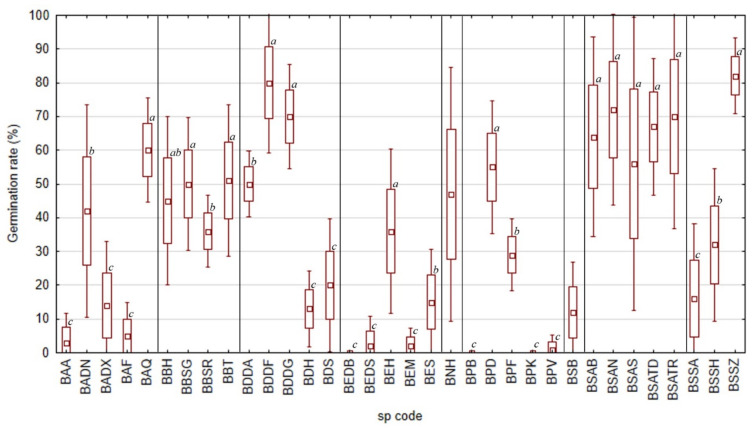
Seed germination rate of all nine taxa of *Boswellia* from different localities in 2021 on Socotra Island (a, b, c—Indexes denoting localities with significantly separate germination rates within individual taxon, the same index associates the seed without statistically significant difference in germination rate).

**Figure 4 plants-11-01418-f004:**
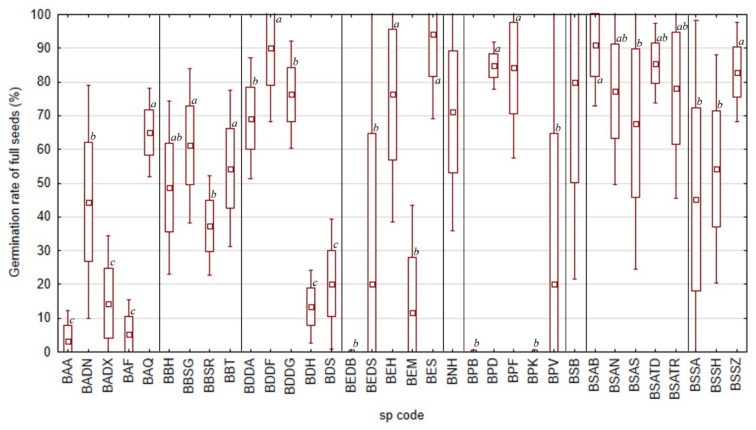
Seed germination rate of full seeds of all nine taxa of *Boswellia* from Socotra Island from different localities in 2021 (a, b, c—Indexes denoting localities with significantly separate germination rates within individual taxon, the same index associates the seed without statistically significant difference in germination rate of full seeds).

**Figure 5 plants-11-01418-f005:**
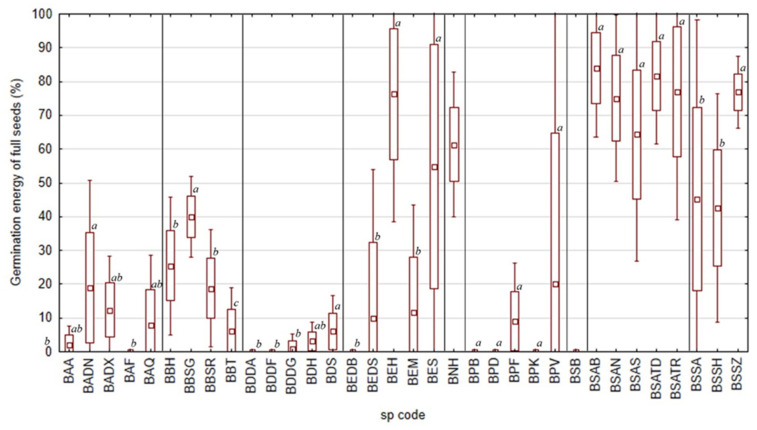
Seed germination energy of full seeds of all nine taxa of *Boswellia* from different localities in 2021 (a, b, c—Indexes denoting localities with significantly separate germination rates within individual taxon, the same index associates the seed without statistically significant difference in germination energy of full seeds).

**Figure 6 plants-11-01418-f006:**
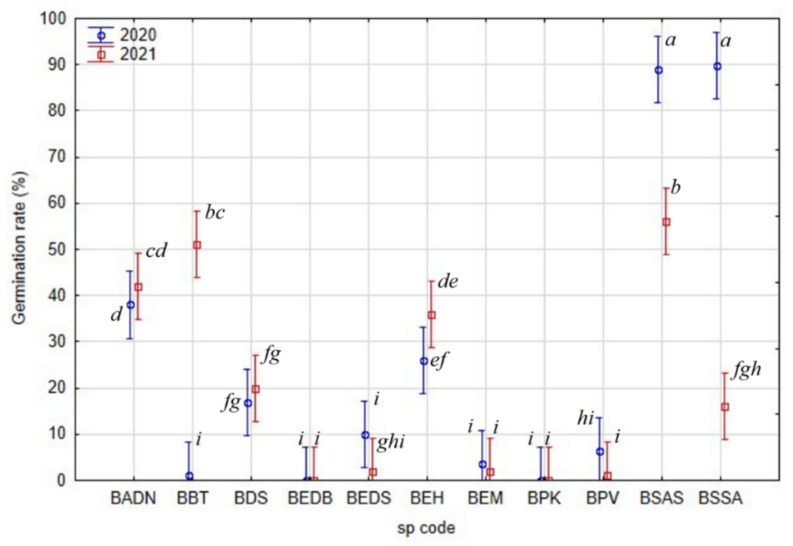
Inter-annual variability of seed germination rate of tested *Boswellia* species (a, b, c, d, e, f, g, h, i—Indexes denoting localities with significantly separate germination rates within individual taxon, the same index associates the seed without statistically significant difference in germination rate).

**Figure 7 plants-11-01418-f007:**
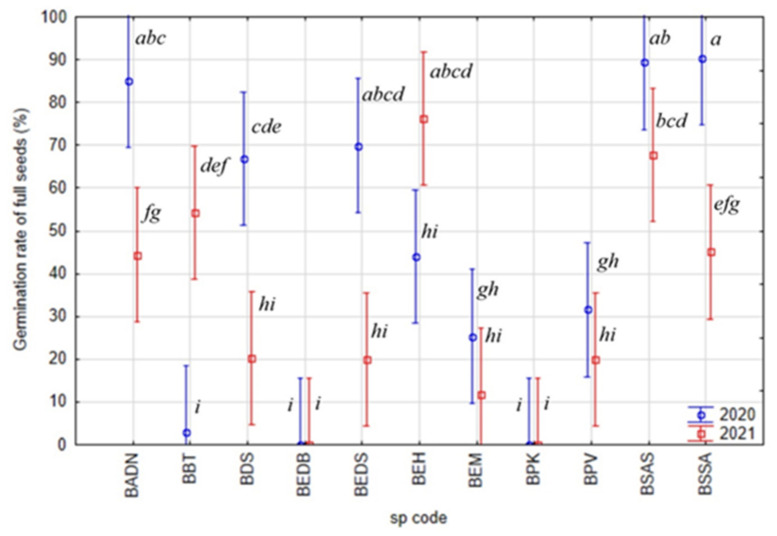
Inter-annual variability of germination rate of full seeds of tested *Boswellia* species (a, b, c, d, e, f, g, h, i—Indexes denoting localities with significantly separate germination rates within individual taxon, the same index associates the seed without statistically significant difference in germination energy of full seeds).

**Figure 8 plants-11-01418-f008:**
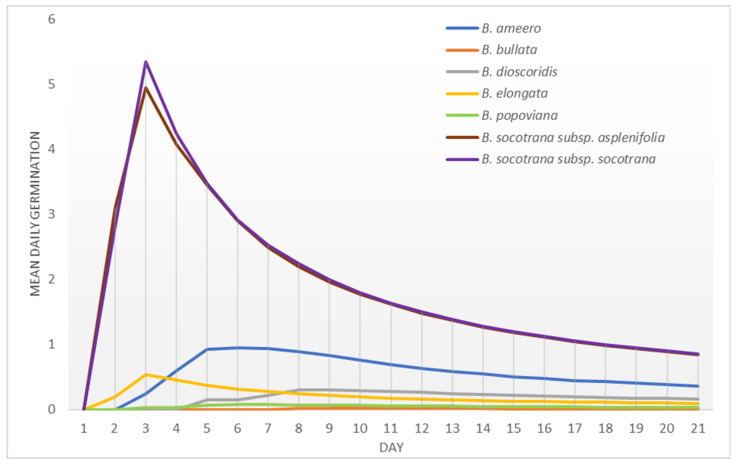
Peak value indexes of seeds of different *Boswellia* taxa from Socotra sampled in 2020.

**Figure 9 plants-11-01418-f009:**
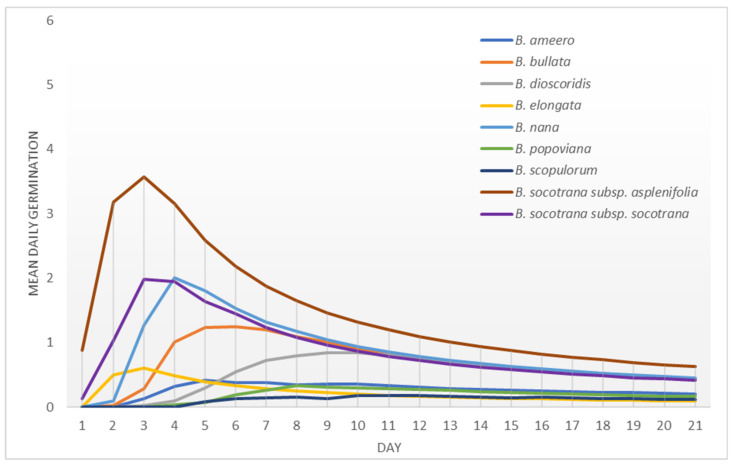
Peak value indexes of seeds of different *Boswellia* taxa from Socotra sampled in 2021.

**Figure 10 plants-11-01418-f010:**
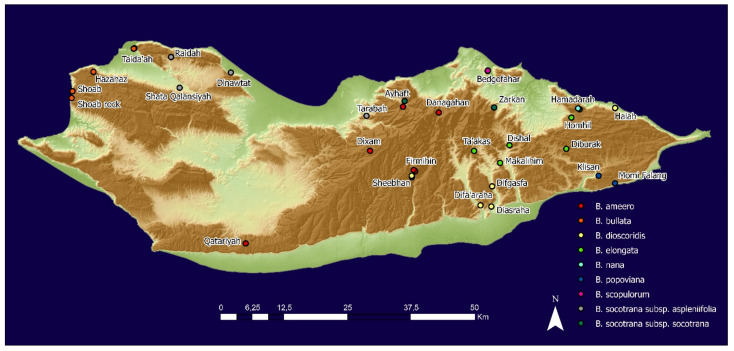
Map of the study area (Socotra Island, Yemen) showing the locations where seeds of *Boswellia* species were collected.

**Figure 11 plants-11-01418-f011:**
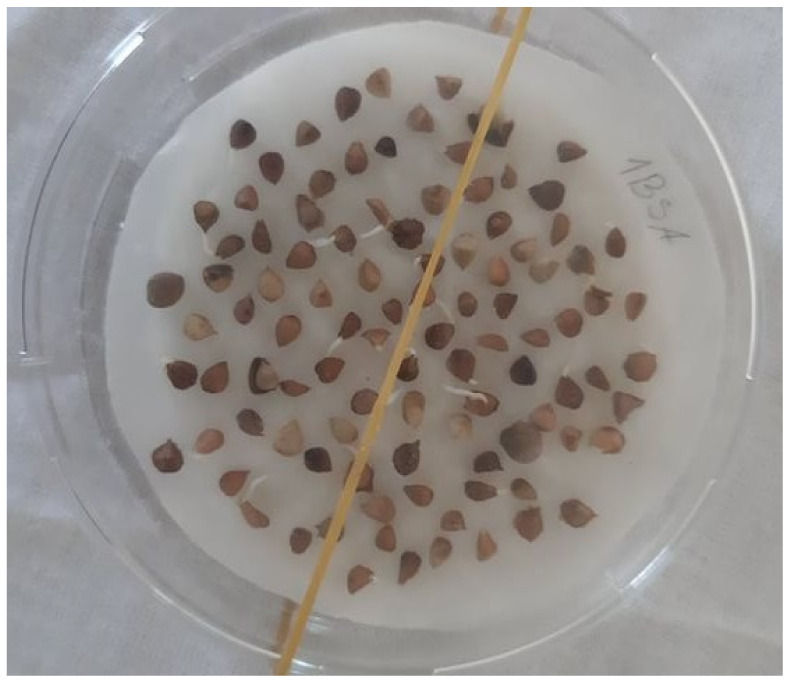
Petri dish with seeds of *Boswellia socotrana* subsp. *aspleniifolia*, some of them already developed radicle longer than the seeds, i.e., they are deemed as germinated.

**Table 1 plants-11-01418-t001:** List of codes, (sub)species, locality names, coordinates, altitude, and date of collection of the 9 Boswellia taxa from Socotra Island (Yemen) assessed in this study.

Species	Locality	Sp. Code	Coordinates	Altitude	Date of Collection
*Boswellia ameero* (BA)	Ayhaft	BAA	12.594013 N 53.988233 E	424	18 May 2021
	Danagahan	BADN	12.585567 N 54.052267 E	629	2 June 2020, 27 May 2021
	Dixam	BADX	12.517885 N 53.930242 E	928	17 May 2021
	Firmihin	BAF	12.481588 N 54.010128 E	617	17 May 2021
	Qatariyah	BAQ	12.351153 N 53.708817 E	860	1 June 2021
*Boswellia bullata* (BB)	Hazahaz	BBH	12.657705 N 53.440213 E	476	24 May 2021
	Shoab rock (green flowers)	BBSG	12.62341 N 53.403162 E	1	7 May 2021
	Shoab (red flower)	BBSR	12.61131 N 53.401565 E	1	7 May 2021
	Taida’ah (Ditwah mountain)	BBT	12.697685 N 53.514572 E	215	20 June 2020, 15 May 2021
*Boswellia* *dioscoridis* (BD)	Wadi Diasraha	BDDA	12.419418 N 54.144430 E	67	26 May 2021
	Wadi Difa’araha	BDDF	12.423367 N 54.123345 E	78	26 May 2021
	Wadi Digasfa	BDDG	12.455117 N 54.146743 E	167	26 May 2021
	Halah (Hoq cave)	BDH	12.591258 N 54.360668 E	70	27 May 2021
	Sheebhan	BDS	12.473968 N 54.004037 E	572	20 May 2020, 17 May 2021
*Boswellia* *elongata* (BE)	Diburak	BEDB	12.523304 N 54.277032 E	360	7 June 2020, 3 June 2021
	Dishal	BEDS	12.527823 N 51.177206 E	282	26 May 2020, 2 June 2021
	Homhil	BEH	12.576047 N 54.286267 E	335	1 June 2020, 10 June 2021
	Makalihim	BEM	12.505041 N 54.168744 E	169	28 May 2020, 2 June 2021
	Shabarah	BES	12.524795 N 53.885643 E	441	4 July 2021
*Boswellia nana* (BN)	Hamadarah (Homhil)	BNH	12.593068 N 54.303272 E	616	19 May 2021
*Boswellia* *popoviana* (BP)	Bedgofahar	BPB	12.657987 N 54.166502 E	178	30 May 2021
	Sheebhan (Didrafarantan)	BPD	12.471406 N 54.004312 E	570	5 June 2021
	Firmihin	BPF	12.482612 N 54.009487 E	608	17 May 2021
	Klisan	BPK	12.473802 N 54.335207 E	270	19 June 2020, 10 June 2021
	Momi Falang	BPV	12.460495 N 54.364178 E	304	19 June 2020, 10 June 2021
*Boswellia* *scopulorum* (BS)	Bedgofahar	BSB	12.656790 N 54.139097 E	149	20 May 2021
*Boswellia* *socotrana* subsp. *aspleniifolia* (BSA)	Raidah (Taida’ah)	BSAB	12.6832650 N 53.5772433 E	346	6 June 2021
	Dinawtat	BSAN	12.655353 N 53.683472 E	34	16 May 2021
	Shata Qalansiyah	BSAS	12.629508 N 53.592948 E	128	28 May 2020, 17 May 2021
	Taida’ah (Ditwah mountain)	BSATD	12.699898 N 53.515875 E	288	15 May 2021
	Tarabah	BSATR	12.579373 N 53.924450 E	128	9 June 2021
*Boswellia* *socotrana* subsp. *socotrana* (BSS)	Ayhaft	BSSA	12.596872 N 53.989797 E	242	20 May 2020, 23 June 2021
	Hamadarah (Homhil)	BSSH	12.591187 N 54.301247 E	466	12 June 2021
	Zarkan (Asmin)	BSSZ	12.595920 N 54.151198 E	408	15 June 2021

## Data Availability

Data used in this article are available from the authors.

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
