# Peer review of "Seed Viability and Potential Germination Rate of Nine Endemic Boswellia Taxa (Burseraceae) from Socotra Island (Yemen)"

_plants, 2022, doi:10.3390/plants11111418_

Round 1

Reviewer 1 Report

Minor Revisions

Review Comments:

Overall comments:

The present study presents for Seed Viability and Potential Germination Rate of Nine Endemic Boswellia Taxa.

The methods are clearly described, the data is clearly presented and defended by the data in the results and discussion. The statistical evaluation is very clear.

Originality

Originality is OK. The idea is sound, and the data obtained is original and useful to understanding the future chances of success of local conservation endeavours of Endemic Boswellia Taxa.

Methodology

The methodology is analytical and there is detail on what was done and how it was statistically analysed. There are 3 points that could be clarified.

Clarity of Presentation

The presentation has a good hypothesis and objective, and the paper is easy to read and to follow to logical conclusions. There is a moderate to extensive reference list, but it is useful (it contains recent papers).

Potential Significance

The potential significance is very good.

Overall:

Overall, I think this manuscript discusses very useful data and has a good objective and hypothesis. It needs minor revision mainly to text editing to be presented as a manuscript in print in Plants.

Corrections to text editing

  1. 30: please use alphabetical order for keywords

Results:

Are there any photos of germinated seeds?

Material and Methods

  1. 384: where did germination take place? In Petri dishes? You should describe clearly
  2. 390: is there any reference on similar species? According to International Seed Testing Association, germination starts when a radicle of 2 mm is formed.

Figures:

Figure 1, 2, 8, 9. Write units on axis yy

I suggest you could use another colour instead of red. A 8% of people do not discriminate red and green colour (Colour_blind_disambiguation)

Author Response

Corrections to text editing

  1. 30: please use alphabetical order for keywords

Response: fixed

Results:

Are there any photos of germinated seeds?

Response: yes, the photo was added to the article as a figure 11

Material and Methods

  1. 384: where did germination take place? In Petri dishes? You should describe clearly

Response: All germination trials were carried out in Socotra in Petri dishes

  1. 390: is there any reference on similar species? According to International Seed Testing Association, germination starts when a radicle of 2 mm is formed.

Response: Boswellia is not in ISTA. We used the same methodology when determining a germinated seed as in Eshete et al. 2012 (citation 16).

Figures:

Figure 1, 2, 8, 9. Write units on axis yy

I suggest you could use another colour instead of red. A 8% of people do not discriminate red and green colour (Colour_blind_disambiguation)

Response: Color red changed to vermillion which should be distinguishable from green in case of colour blindness (https://media.springernature.com/lw685/springer-static/image/art%3A10.1038%2Fnmeth.1618/MediaObjects/41592_2011_Article_BFnmeth1618_Fig2_HTML.jpg?as=webp)

Reviewer 2 Report

In this paper the authors showed germination results for different Boswellia species. The methods used in this study are vary basic, however the authors carefully analyzed all data with statistical test. The results seems to be useful for conservation programs.

However in my opinion the results are veery general. In my opinion some more advances studies could be proposed in course of realization of this project.

If the plants from Boswellia species are classify as endangered the authors should add information about agreements with local governments to collect and use of the seeds from this species for study purposes. There is lack of such information in the manuscript.

The English style is fine.

Author Response

In this paper the authors showed germination results for different Boswellia species. The methods used in this study are vary basic, however the authors carefully analyzed all data with statistical test. The results seems to be useful for conservation programs.

However in my opinion the results are veery general. In my opinion some more advances studies could be proposed in course of realization of this project.

Response: We agree, this study is a first ever study focusing on the germination in such a wide range of endemic taxa and localities of Boswellia on Socotra Island and is the basis for further studies that our team will carry out. So in the future, more advanced studies will be used.

If the plants from Boswellia species are classify as endangered the authors should add information about agreements with local governments to collect and use of the seeds from this species for study purposes. There is lack of such information in the manuscript.

Response: All the germination trials were carried out in Socotra, and permission from local authorities is required  only in case of export of plant material.  Sentence “All seed germination tests were carried out in Socotra.” was added to the text in Materials and Methods

Reviewer 3 Report

The authors of this manuscript examined the germination of several endemic Boswellia species (Burseraceae) of Socotra Island (Yemen) which are of great local and global significance due to their various local ethnobotanical uses. The authors have collected seeds for 9 taxa island-wide from a wide range of localities, including several populations per studied species, and they have tested seed germination in controlled experiments for two subsequent years. This study sheds light on the problem of poor regeneration of Boswellia species on Socotra Island reporting that such a trend is not caused by internal biological factors related to generative reproduction but rather is connected with external ecological factors acting on the growth of seedlings after germination. Being a conservation-oriented study, the findings of this manuscript offer novel insight into the Socotran terrestrial ecosystems and their unique floristic elements and can also aid the conservation efforts of local communities and authorities.

Although this study is quite interesting and merits consideration, some improvements should clearly be made prior to publication. Some general issues are outlined below:

  • The abstract can be enlarged to include more insight into the results obtained.
  • The authors should add the authorship of the scientific names they refer to at their first mention in the texts either from the recent monograph of the genus (see book presentation at DOI: 10.36253/jopt-9299) or from the Plant List or World Flora Online.
  • The authors should state clearly in section 4.1 that they have contacted the Environmental Protection Authority (Hadibo, Socotra) for permission to access the wild-growing populations to implement their research (Prior Informed Consent in terms of the Nagoya Protocol). If they have written permission as a document, they should also refer to the granted access explicitly.
  • The authors should provide a reference for the chosen conditions for seed germination.
  • Section 4.3 should be elaborated and presented as a coherent text and not as handouts or bullet-point notes.
  • The letter p in significance levels should be italicized throughout the manuscript and legends.
  • The abbreviation circa should be abbreviated as ‘ca.’ (and not as ca); this should be corrected throughout the manuscript.

Although the manuscript is clearly written and well-structured, there are many details in various parts that need the authors’ attention. Several linguistic imperfections should be eliminated or some sentences should be rephrased to reach high language standards. Below these cases are indicated per the basic manuscript’s section:

  1. Introduction

Line 20: Delete comma

Line 25-26-27: noted (instead of note)…. showed (instead of show) … indicated (instead of indicates)… tree taxa (instead of Tree taxa)

Lines 59-60: Delete “in comparison to now”

Line 62-63: Use present tense, not past tense and connect to the reference cited.

Line 66: The word ‘and’ between the two Latin names should not be italicized.

Line 76: …of these species (instead of …of the species)

Line 80: … such as those in Socotra Island (and not such as in Socotra Island)

Line 83: Add family and authorship for Dracaena cinnabari.

Line 88: …tree taxa (instead of Tree taxa)

Line 92: Delete ‘by’ at the end of the sentence.

Line 100: [16-19].

  1. RESULTS

Line 114: … of Different Boswellia Taxa … (instead of …of Different Taxa of Boswellia)

Line 116: socotrana – the last two letters should also be italicized.

Line 118: Boswellia (and not Boswelia)

Line 120 and 121 and 124 and 126 and 136 and 137 and 142 (and many more in sections 2.1 and 3.1): No space is needed between the numbers and %.

Line 123: B. elongata should be italicized.

Lines 137-142: Seeds of B. bullata contained ca 35-50 % germinated and fresh seeds at all localities; seeds of other species reached a similar proportion of these seeds only at one locality, i.e., B. ameero (BADN) from Danagahan, B. elongata (BEH) from Homhil, B. nana (BNH) from Hamadarah and B. popoviana (BPD) from Sheebhan); the seeds from other localities contained the minimal proportion of these seeds (usually up to 15%) and showed significantly lower germination rate.

Line 134: Fig. (not Fig)

Line 159: B. elongata should be italicized.

Lines 205 and 210: a, b, c, d, e, f, g, h, i – Add space after the comma used, and add a comma between e and f.

Lines 206-207 and 210-211: The wording ‘within individual taxon’ is rather problematic; please consider rephrasing.

Line 216: Add a comma before the word and (when referring to B. popoviana and B. scopulorum).

Line 217: Delete the word ‘and’ before B. socotrana.

Line 219: Add ‘while’ before ‘the remaining’.

Line 222: were found (add were)

Line 220: In 2020 and 2021, (add a comma).

Line 254: Add ‘Nevertheless,’ at the beginning of the sentence.

Line 256: Replace ‘however’ with ‘nonetheless’.

Line 270: … in detail by Raju et al. (25] and Sunnichan et al. [26], respectively.

Line 271: above-mentioned

Line 276: did not (instead of didn´t)

Line 278: … report (not mentioned) mainly bees as principal insect pollinators.

Line 281: …garden lizards as predators of pollinating insects…

Line 282: which (instead of that)

  1. DISCUSSION

Line: 235: sp. prov. (instead of spec. prov.)

Line 238: six (not 6)

Line 241: report (not recorded)

Lines 247-248: Last investigated species was Boswellia dalzielii [17], associated with a germination rate in the field experiment….

Line 297: with the exception of BEDB and BPK.

Line 299: Add the (the problem)

Lines 305-306: Delete comma after B. dioscoridis.

Line 308: Add a comma before the word ‘and’.

Line 312: from (not since)

Line 321: start (not started)

Lines 324-326: Swartout and Solowey [14] even report lack of dormancy in Boswellia species which means that seeds of Boswellia species do not actually need any pre-sowing treatment; the best solution is to soak them in cold water to germinate [20], as they likely cannot be stored at room temperatures indefinitely.

Line 330: the case study

Line 332: the entire area

Line 337: This (instead of It)

Lines 339-343: All the above-mentioned authors have listed several factors to explain the missing of young age classes of trees in Boswellia populations, more often intensive resin harvesting leading to extensive stem injuries and to decreasing of trees vitality that can be thereafter more easily affected by insect and fungal diseases; frequent human-induced fires; changes in land-use followed by land degradation or over-grazing.

Line 346-347: On the other hand, some authors mention the aspect of inaccessibility in species growing in rocky habitats (rock-dwellers).

Lines 351-352: An alternative way of regeneration could be the vegetative propagation, as an adaptation to specific stress conditions such as wildfires or prolonged drought.

Line 354: the ability

Lines 355-360: Propagation by cuttings with a high percentage of rooting has been reported by Abiyu et al. [36] and Haile et al. [37] which may help overcome the population recruitment bottleneck of Boswellia species [37], if properly managed and assisted to enhance the genetic diversity of various genotypes, including the directed propagation of natural hybrids and hybridogenic species that appear very often on Socotra [15,38-39] due to common occurrence and proximity of ten different species with distinct characteristics.

  1. MATERIALS AND METHODS

Line 380: Delete comma after (Socotra Island, Yemen).

Line 383: Delete comma after (Yemen).

  1. CONCLUSIONS

Line 456: …tree species (instead of Tree species).

Line 460: improves (instead of betters).

Line 368: Check formatting.

  1. APPENDIX

Appendix A, Tables 2 and 3: Use Boswellia species (instead of Species) as a header in the first column and refer to the number of studied species in their legends (e.g., … of different Boswellia taxa of Socotra sampled…).

  1. REFERENCES

Line 491: Use the abbreviation for the Journal of the Drylands.

Line 507: One or two taxa? (instead of One or Two Taxa?)

Line 525-526: Boswellia ovalifoliolata (in italics)

Lines 527-528: Causes of low seed quality in Ilex paraguariensis A. St. Hil. samples (Aquifoliaceae) – No word capitalization is needed.

Line 567: Frankincense revisited, part II: Volatiles in rare Boswellia species and hybrids – No word capitalization is needed.

Line 562: Forest Sci. (instead of Forest science)

Line 572: Use the abbreviation for the Journal Rendiconti Lincei Scienze Fisiche e Naturali.

Author Response

The abstract can be enlarged to include more insight into the results obtained.

Response: Done

The authors should add the authorship of the scientific names they refer to at their first mention in the texts either from the recent monograph of the genus (see book presentation at DOI: 10.36253/jopt-9299) or from the Plant List or World Flora Online.

Response: Done

The authors should state clearly in section 4.1 that they have contacted the Environmental Protection Authority (Hadibo, Socotra) for permission to access the wild-growing populations to implement their research (Prior Informed Consent in terms of the Nagoya Protocol). If they have written permission as a document, they should also refer to the granted access explicitly.

Response: Done

The authors should provide a reference for the chosen conditions for seed germination.

Response: Done

Section 4.3 should be elaborated and presented as a coherent text and not as handouts or bullet-point notes.

Response: Done

The letter p in significance levels should be italicized throughout the manuscript and legends.

Response: Done

The abbreviation circa should be abbreviated as ‘ca.’ (and not as ca); this should be corrected throughout the manuscript.

Response: Done

Although the manuscript is clearly written and well-structured, there are many details in various parts that need the authors’ attention. Several linguistic imperfections should be eliminated or some sentences should be rephrased to reach high language standards. Below these cases are indicated per the basic manuscript’s section:

Response: changes were carried out according to suggestion of the reviewer in the whole manuscript